

# Unraveling the physiological responses of morphologically distinct corals to low oxygen

Ying Long[1,2], Sutinee Sinutok[1,3], Pimchanok Buapet[1,4] and Mathinee Yucharoen[1,2]

[1] Coastal Oceanography and Climate Change Research Center, Prince of Songkla University, Hat Yai, Songkhla, Thailand
[2] Marine and Coastal Resources Institute, Faculty of Environmental Management, Prince of Songkla University, Hat Yai, Songkhla, Thailand
[3] Faculty of Environmental Management, Prince of Songkla University, Hat Yai, Songkhla, Thailand
[4] Division of Biological Science, Faculty of Science, Prince of Songkla University, Hat Yai, Songkhla, Thailand

Corresponding author
Mathinee Yucharoen,
mathinee.y@psu.ac.th

## ABSTRACT

**Background**. Low oxygen in marine environments, intensified by climate change and local pollution, poses a substantial threat to global marine ecosystems, especially impacting vulnerable coral reefs and causing metabolic crises and bleaching-induced mortality. Yet, our understanding of the potential impacts in tropical regions is incomplete. Furthermore, uncertainty surrounds the physiological responses of corals to hypoxia and anoxia conditions.

**Methods**. We initially monitored *in situ* dissolved oxygen (DO) levels at Kham Island in the lower Gulf of Thailand. Subsequently, we conducted a 72-hour experimental exposure of corals with different morphologies—*Pocillopora acuta*, *Porites lutea*, and *Turbinaria mesenterina*—to low oxygen conditions, while following a 12/12-hour dark/light cycle. Three distinct DO conditions were employed: ambient (DO $6.0 \pm 0.5$ mg L$^{-1}$), hypoxia (DO $2.0 \pm 0.5$ mg L$^{-1}$), and anoxia (DO $< 0.5$ mg L$^{-1}$). We measured and compared photosynthetic efficiency, Symbiodiniaceae density, chlorophyll concentration, respiratory rates, primary production, and calcification across the various treatments.

**Results**. Persistent hypoxia was observed at the study site. Subsequent experiments revealed that low oxygen levels led to a notable decrease in the maximum quantum yield over time in all the species tested, accompanied by declining rates of respiration and calcification. Our findings reveal the sensitivity of corals to both hypoxia and anoxia, particularly affecting processes crucial to energy balance and structural integrity. Notably, *P. lutea* and *T. mesenterina* exhibited no mortality over the 72-hour period under hypoxia and anoxia conditions, while *P. acuta*, exposed to anoxia, experienced mortality with tissue loss within 24 hours. This study underscores species-specific variations in susceptibility associated with different morphologies under low oxygen conditions. The results demonstrate the substantial impact of deoxygenation on coral growth and health, with the compounded challenges of climate change and coastal pollution exacerbating oxygen availability, leading to increasingly significant implications for coral ecosystems.

# INTRODUCTION

Oceans worldwide are experiencing a decline in oxygen levels as the climate warms and coastal pollution accelerates, which could have adverse effects on the diversity and richness of marine organisms (*Bopp et al., 2013*; *Breitburg et al., 2018*; *Camp et al., 2018*; *Sampaio et al., 2021*). 'Hypoxia' is defined by oxygen levels less than 2 mg L$^{-1}$, and this is a condition that some studies have suggested may impose more severe impacts on marine life than ocean warming, ocean acidification, or their combined effects (*Vaquer-Sunyer & Duarte, 2008*; *Bijma et al., 2013*; *Haas et al., 2014*).

According to the Intergovernmental Panel on Climate Change (IPCC) and their representative concentration pathways (RCP 2.6 and 8.5), the dissolved oxygen (DO) content is projected to decrease by between 1.7% and 4% by 2100 due to climate change drivers (*IPCC, 2022*). Over the past 50 years, certain tropical areas, including the Central Pacific and the Indian Ocean, have experienced a significant decline, with up to a 40% reduction in their DO levels (*Schmidtko, Stramma & Visbeck, 2017*). This decrease is primarily attributed to the absorption of rising atmospheric $CO_2$ from human activities and the impact of consequent excessive atmospheric heat (*Levin & Bris, 2015*; *Henson, Beaulieu & Lampitt, 2016*). As the oceans warm, the solubility of oxygen in seawater decreases, and simultaneously, the physiological oxygen requirements for many organisms increase (*Lucey et al., 2023*). This scenario can lead to altered behavior, migrations, decreased growth rates, reduced fecundity, and higher mortality rates (*Levin & Bris, 2015*; *Breitburg et al., 2018*). In addition, coastal areas are experiencing hypoxic or anoxic conditions due to factors such as, eutrophication and restricted circulation (*Nakamura, Yamasaki & Van Woesik, 2003*; *Ulstrup, Hill & Ralph, 2005*; *Keeling, Körtzinger & Gruber, 2009*).

Hermatypic scleractinian corals are pivotal as the primary reef-building species, thriving in shallow, warm water environments with adequate light. They play a vital role in supporting a diverse array of marine species by providing food, shelter, and substrate (*Liao, Xiao & Li, 2019*; *Raphael et al., 2020*). Their metabolic needs, constituting up to 90% of metabolism, are fulfilled through a mutualistic interaction with endosymbiotic dinoflagellate algae known as Symbiodiniaceae (*Muscatine, 1990*). However, unfavorable environmental factors can lead to the disruption of this essential symbiosis (*Zhu et al., 2004*; *Suggett & Smith, 2020*), with hypoxia acknowledged as one of the primary drivers.

The extent and consequences of low oxygen are increasingly recognized (*Hughes et al., 2020*). Previous findings underscore a growing concern as they highlight the widespread deaths of corals and coral reef associated animals attributed to hypoxia and dead zones (*Altieri & Gedan, 2015*; *Altieri et al., 2017*). Notably, the consequences of coral mass mortality extend beyond direct impacts, as many faunas associated with coral reef habitats are also affected (*Diaz & Rosenberg, 2008*; *Galic, Hawkins & Forbes, 2019*; *Raj et al., 2020*; *Alderdice et al., 2021*). It has been established that inadequate oxygen hampers cellular

processes, deteriorating coral health and rendering it susceptible to severe bleaching under hypoxia (*Alderdice et al., 2021*; *Figuerola et al., 2021*; *Jain et al., 2023*). Our recent study along the Andaman coast of Thailand reveals that hypoxia significantly impacts various coral health parameters, resulting in reduced photosynthetic efficiency, Symbiodiniaceae density, chlorophyll concentration, and overall coral growth in certain species (*Jain et al., 2023*). The study further emphasizes distinct susceptibility levels to hypoxia among the different tested coral species, underscoring the importance of identifying species-specific responses for effective management strategies (*Jain et al., 2023*).

The previous work by Kham Island in the southern Gulf of Thailand was initiated by measuring *in situ* DO. Building upon these data, an experimental approach was employed to assess the susceptibility to low oxygen conditions among three morphologically distinct dominant coral species at Kham Island: *Pocillopora acuta*, *Porites lutea*, and *Turbinaria mesenterina*. The investigation explored changes in the physiological performances and metabolism of these corals across a range of DO levels categorized as hypoxia and anoxia. As the first findings from the lower Gulf of Thailand and complementing our previous study, this research aims to offer guidance for prioritizing management initiatives to alleviate the adverse effects of low oxygen in tropical shallow-water coral reefs. Within the broader context of global climate change, the study provides essential baseline information to enhance ecological risk assessment.

## MATERIALS & METHODS

### Assessment of *in situ* environmental parameters

The study site is located at the western part of Kham Island (6°58′24.3″N 100°51′24.8″E), situated in the lower Gulf of Thailand within Songkhla Province (Fig. 1). The depth in the study area ranges within 3–5 m. According to the Department of Marine and Coastal Resources, Thailand (DMCR) survey conducted in 2019, the primary reef areas in both the northern and southern regions of the island were reported to be in very good condition.

To establish baseline conditions for experimental simulations, we recorded environmental parameters at the study site. One HOBO® U26-001 data logger (Onset, USA) was strategically positioned at a depth of 5 m, situated adjacent to the coral colonies 0.5 m within the western reef of Kham Island at the same site from where the corals were sampled. This logger was programmed to record DO values at hourly intervals from June 2021 to June 2022 (except for January 2022 to March 2022), contributing to a detailed temporal profile of the DO dynamics within the specified aquatic environment. Additionally, we employed the AAQ-RINKO 176 multiprobe (JFE Advantech Co. Ltd., Hyogo, Japan) to collect data on various parameters, including temperature, salinity, chlorophyll *a* concentration, pH and irradiance monthly. Detailed information is provided in Table S1.

### Coral sampling and acclimation

In June 2022, healthy colonies ($n = 8$) of each of the coral species *P. acuta, P. lutea* and *T. mesenterina*, representing three different morphological corals were collected using stainless hammer and chisel. These are the dominant coral species at Kham Island (*Department of*
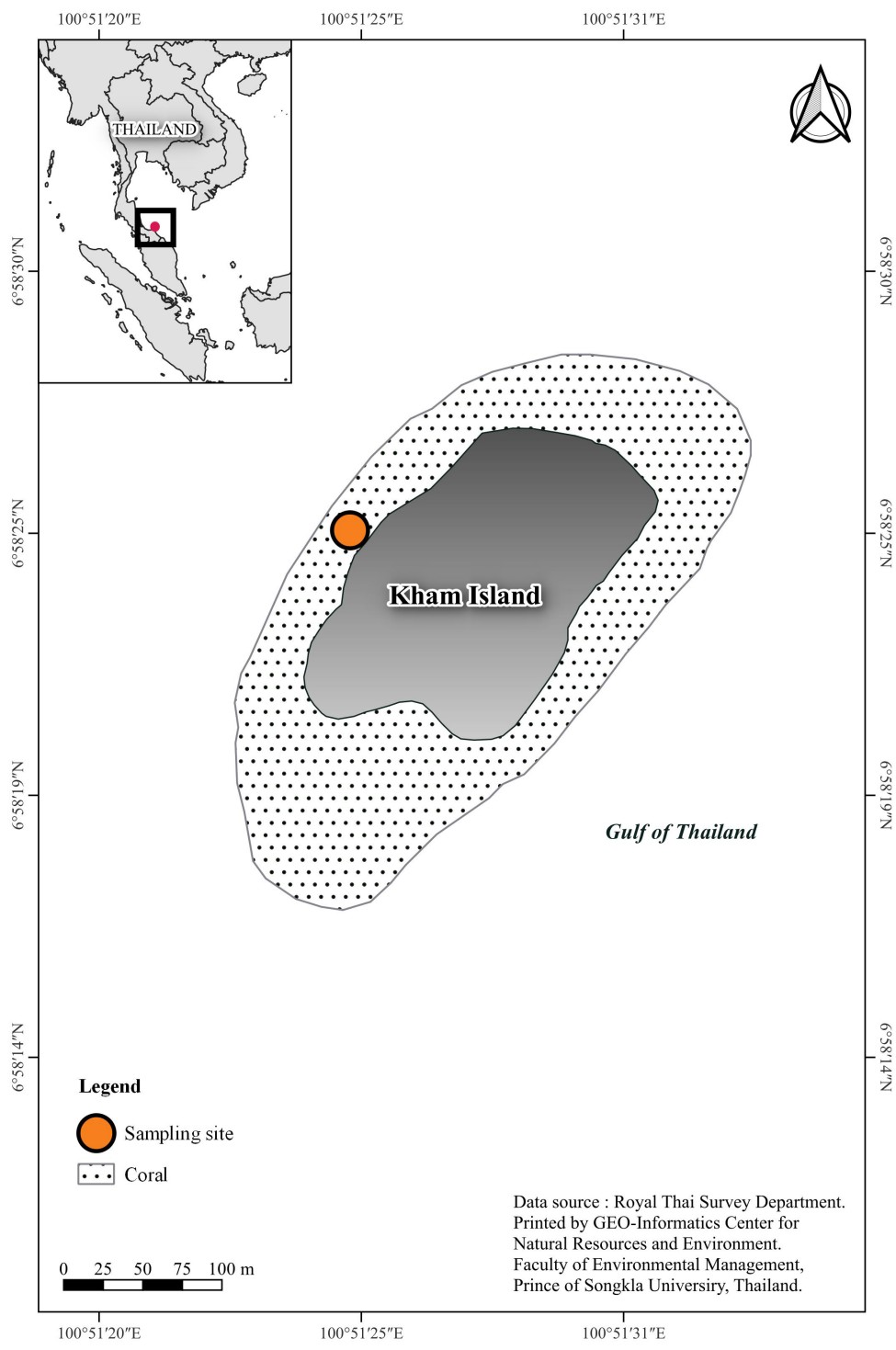

**Figure 1** **Sampling site location, southern Gulf of Thailand.** Coral basemap is modified from topographic map scales 1:50,000 of Royal Thai Survey Department by Geo-Informatics Center for Natural Resources and Environment, Prince of Songkla University.



*Marine and Coastal Resources, 2019*). The research permission in the Non-Hunting Area was approved by the Department of National Parks, Wildlife and Plant Conservation (permission number: 21685). Coral collection was permitted by the Department of Fisheries, Ministry of Agriculture and Cooperatives (permission number: 409) under Wild Animal Conservation and Protection Act, B.E. 2562 (A.D. 2019). The live samples were transferred to the aquarium facility of Coastal Oceanography and Climate Change Research Center (COCC) at Prince of Songkla University (PSU) within 2 h. Here, they were acclimated in a 600 L holding tank that simulated the environmental conditions (light 120 $\mu$mol photons m$^{-2}$ s$^{-1}$, temperature 29 °C, salinity 32 psu, and pH 8.20) of the sampling area. Throughout the acclimation process, the maximum quantum yield (MQY) of the coral was evaluated on a daily basis to evaluate its recovery and overall health after being fragmented.

## Experiment design

The experiment of this study was conducted according to the Animals for Scientific Purposes Act, B.E. 2558 (A.D. 2015) and approved by Institutional Animal Care and Use Committee, Prince of Songkla University (ref.46/2021). Each coral colony was cut into four nubbins, each measuring 3–5 cm. A total of 96 coral nubbins (32 per species) were cut from 24 source colonies (eight per species). After additional acclimation, we selected 24 nubbins from the aforementioned colonies (eight per species) to assess their initial physiological status through Symbiodiniaceae density and chlorophyll concentration analysis. The remaining 72 nubbins (24 per species) were then subjected to treatment conditions (one nubbin per colony per treatment), with each treatment or species having eight replicates ($n = 8$). This ensured coverage across eight distinct genetic colonies. To maintain experimental integrity, each nubbin was individually housed in a closed chamber with a volume of 710 cm$^3$. The treatments were as follows: (1) Ambient with DO levels ranging from 6.0 to 6.5 mg L$^{-1}$, (2) Hypoxia with DO levels ranging from 1.5 to 2.5 mg L$^{-1}$, and (3) Anoxia with DO levels ranging from 0 to 0.5 mg L$^{-1}$ (refer to Fig. 2 for graphical representation). DO levels were adjusted in 50 L stock seawater tanks using a nitrogen high-pressure regulator (IM-TCH, China) with an air compressor pressure regulator (Xcpc, China), meanwhile, we closely monitored pH ensuring a difference within approximately 0.1 across treatments. Prepared seawater was added to each chamber, and coral nubbins were gently placed inside. All chambers were then sealed with parafilm (Bemis, Sheboygan Falls, WI, USA).

The experiment ran for 72 h in a 12:12 dark/light cycle, commencing in a dark condition. Throughout the experiment, light, temperature, and salinity were controlled and maintained at the same conditions as during the acclimation period. The seawater in each chamber underwent striation every 3 h and renewal every 12 h, synchronized with the light cycle, utilizing freshly prepared seawater specific to the treatment. DO (mg L$^{-1}$) and pH (NBS scale) measurements were recorded before and after each 12-hour incubation period, alongside simultaneous collection of water samples in each chamber for total alkalinity (mg L$^{-1}$) measurement. The MQY of all nubbins was assessed at the start and end of dark/light conditions (non-destructive sampling). Coral nubbins were

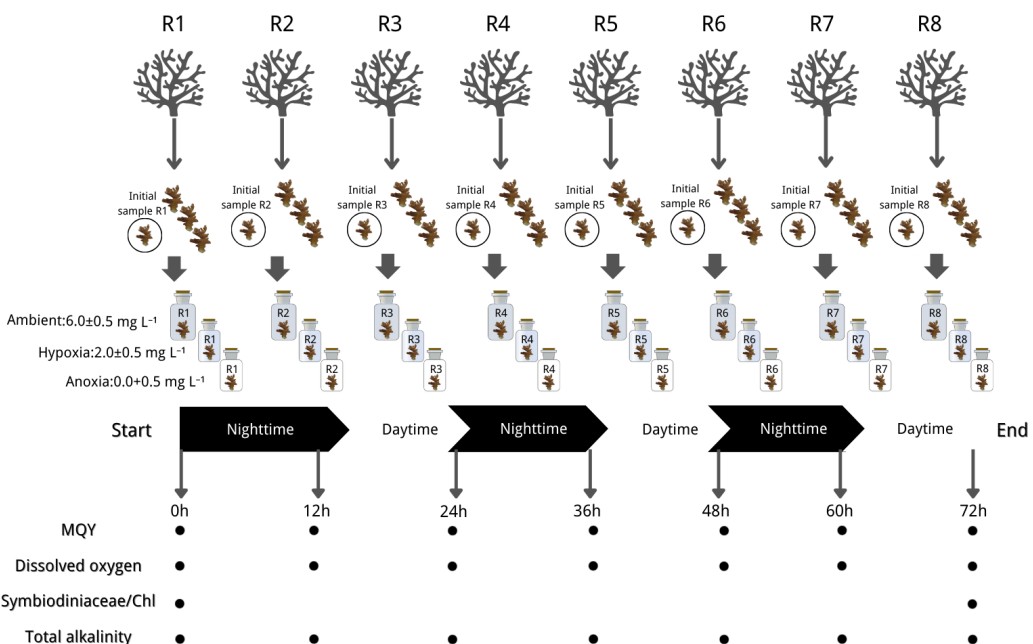

**Figure 2** **Experimental design and sampling parameters.** Three coral species were done in 3 treatments: (1) Ambient with DO levels ranging from 6.0 to 6.5 mg L$^{-1}$, (2) hypoxia with DO levels ranging from 1.5 to 2.5 mg L$^{-1}$, and (3) anoxia with DO levels ranging from 0 to 0.5 mg L$^{-1}$. The experiment ran for 72 h in a 12:12 dark/light cycle with measurements of maximum quantum yield (MQY), dissolved oxygen (DO), Symbiodiniaceae density, chlorophyll content (chl) and total alkalinity. Icon source credit: Canva (the infographic was generated using Canva online software) and coral photos were taken by the first author.

collected at the initial time (three species* eight replicates) and the end of the experiment (three species* eight replicates* three treatments), subsequently stored in a −80 °C liquid nitrogen tank for later analysis of Symbiodiniaceae density and chlorophyll content. The experimental design is summarized in Fig. 2.

## Measurement protocols
### Chlorophyll fluorescence

The photosynthetic efficiency represented by the MQY was evaluated at 9:30 and 22:30 following dark acclimation. Quantification of MQY for *P. acuta*, *P. lutea*, and *T. mesenterina* in each treatment was conducted every 12 h using a Diving-PAM fluorometer (Walz GmbH, Effeltrich, Germany) connected to a 6 mm diameter fiberoptic probe. The PAM settings were held constant with a measuring light intensity (MEAS-INT) of 5, electronic signal gain (GAIN) set to 2, saturation pulse intensity (SAT-INT) at 8, and the width of the saturating light pulse (SATWIDTH) at 0.6 s (*Yucharoen et al., 2021*).

### Symbiodiniaceae density and chlorophyll content

The symbiotic relationship with Symbiodiniaceae was investigated by employing the density of Symbiodiniaceae and chlorophyll content as proxies. Each frozen coral nubbin underwent air-blasting to separate the coral tissue from the skeleton, followed by dissolution in 50 mL artificial seawater (32 psu). The resulting tissue slurries were then
centrifuged at 1,000 rpm for 10 min, and 1 mL of each sample suspension was extracted for Symbiodiniaceae cell counting using a hemocytometer under a light microscope for three technical replicates.

The remaining slurry from cell counting (algal cells and coral tissues) was resuspended in 3 mL of 90% acetone and stored in darkness for 24 h at 4 °C. Subsequently, it was centrifuged at 5,000 rpm for 5 min, and the photosynthetic pigments (chlorophyll *a* and chlorophyll $c_2$) were measured using a spectrophotometer (SP8001; Metertech, Taipei, Taiwan) by taking absorbance readings at 630, 664, and 750 nm. The standard spectrophotometric method described by *Ritchie (2006)* was employed for chlorophyll analysis.

Symbiodiniaceae density and chlorophyll concentration were determined per the surface area of the coral nubbin. The paraffin wax method was utilized to determine the coral's surface area (*Stimson & Kinzie, 1991*).

### Respiration and primary production

DO levels in all chambers were monitored in both dark and light conditions using a multiparameter benchtop meter (inoLab® Multi 9630 IDS; Xylem Analytics, Oberbayern, Germany).

Respiration rate (R) and net primary production (NPP) were subsequently calculated based on the oxygen consumption in dark conditions and the oxygen release in light conditions, respectively. The calculations followed the methodology by *Cohen, Dubinsky & Erez (2016)*. The equation for R (or NPP) is as follows:

$$NPP \text{ or } R \ (\mu g \ O_2 \ cm^{-2} \ h^{-1}) = \frac{(O_2 \ end - O_2 \ start) * (V \ chamber - V \ nubbin)}{Time * Surface \ area}. \tag{1}$$

Here, time is the duration of the measurement (12 h), and the surface area is the surface area of the coral nubbin. This approach allows for estimating the gross primary production as follows:

$$Gross \ primary \ production \ (\mu g \ O_2 \ cm^{-2} \ h^{-1}) = Net \ primary \ production + Respiration \ rate. \tag{2}$$

### Calcification rate

The HI84502 mini titrator (HANNA Instruments, Woonsocket, RI, USA) was employed to perform titrations on seawater for the determination of total alkalinity (TA). As coral calcification is a light-enhanced process (*Mallon et al., 2022*), the change in total alkalinity during light conditions was utilized to calculate the calcification rate, employing the equation outlined by *Cohen, Dubinsky & Erez (2016)*. The equation is as follows:

$$Calcification \ rate \ (\mu mol \ O_2 \ cm^{-2} h^{-1}) = \frac{\frac{\Delta TA}{2} * (V \ chamber - V \ nubbin) * 1000 * 1.028}{Time * Surface \ area}. \tag{3}$$

Here, $\Delta TA$ represents the difference between the initial (TA start) and the final (TA end) total alkalinities, 1.028 is the seawater density (1.028 L × Kg$^{-1}$), the division by 1,000 converts mmol to μmol, time is the duration of the measurement (12 h), and Surface area is the surface area of each coral nubbin. This formula allows for the quantitative assessment of calcification rate per unit surface area over the specified time period and chamber conditions.

## Statistical analysis

All parameters were tested for normality using the Shapiro–Wilk test, with square root or log10 transformations applied as needed for non-normally distributed data. Repeated measures ANOVA was used to evaluate the effects of different oxygen levels on MQY. In this analysis, oxygen level was the categorical factor, while the hours of measurement served as the within-group factor, allowing for the assessment of both treatment effects and temporal changes within the same nubbin. Similarly, repeated measures ANOVA were applied to net primary production, gross primary production, respiration rate, and calcification rate, with oxygen level as the categorical factor and day of measurement as the within-group factor. For *P. acuta*, which experienced tissue loss and mortality within one day of anoxia treatment, a one-way ANOVA was conducted on day 1 data to detect differences in net primary production, gross primary production, respiration rate, and calcification rate among oxygen levels. One-way ANOVA was used to assess significant differences in Symbiodiniaceae density and chlorophyll concentration at the end of the experiment among treatments with different oxygen levels. Statistical significance was determined at a 95% confidence level, with post hoc comparisons performed using the Tukey Honestly Significant Difference (HSD) test.

## RESULTS

### *In-situ* dissolved oxygen (DO)

Between June 2021 and June 2022 (except for 20th January 2022–4th March 2022), DO loggers consistently monitored the DO levels at 12:00 and 24:00 daily in the reef region to the west of Kham Island. The annual average DO concentration, excluding February 2022, was found to be 4.84 mg $L^{-1}$.

Our data suggests the presence of recurring hypoxic events in the reef area of the western part of Kham Island, particularly between September and November 2021. Over this three-month period, the DO in the reef area consistently fell below 4 mg $L^{-1}$, with numerous instances of levels dropping even lower, reaching below 2 mg $L^{-1}$ (refer to Fig. 3). Analyzing the monthly trends, we observed a notable decline in DO levels almost every month from July 2021 through the middle of October 2021. This downward trend continued, with DO levels plummeting to 1 mg $L^{-1}$ in October 2021. Hypoxia was identified during seven out of the 11 recorded months.

### Chlorophyll fluorescence

MQY under ambient treatment of all three species had no discernible impact over time (Table S10), indicating corals in ambient conditions remained in a healthy state. Significantly lower values of MQY were observed in the anoxia treatment compared to the control treatment during nighttime, across all the studied species (Fig. 4). A significant interaction between treatment and time was detected for all three species ($p < 0.001$, see Table S2). In the case of *P. acuta* subjected to anoxia scenario, there was a notable decline in MQY after the initial 12 h in dark condition, leading to a complete loss of coral tissue within 24 h. Similarly, corals exposed to hypoxic conditions exhibited significantly reduced

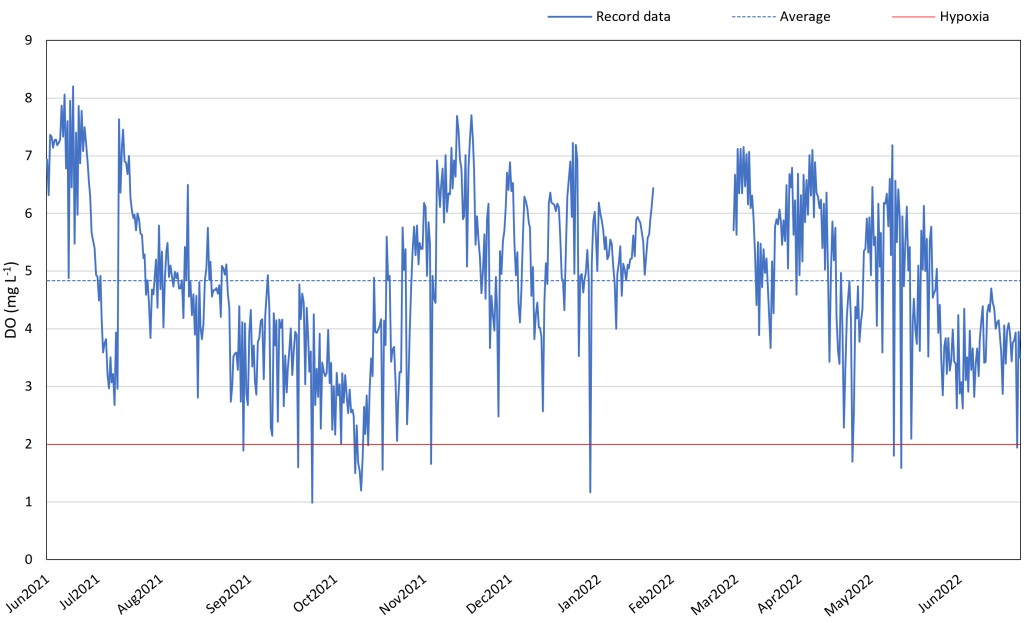

**Figure 3** DO record in coral reef at Kham Island throughout the period from June 2021 to June 2022.

photosynthetic efficiency compared to the control group after a 24 h treatment period
(Fig. 4A), with the effects being less pronounced than those observed under anoxia.

*P. lutea* showed significant differences across treatments as well ($p < 0.001$). In anoxic
condition, corals exhibited significantly decreased photosynthetic efficiency when treated
in the dark compared to the control; however, the efficiency was notably recovered when
the light was on (Fig. 4B).

The performance of *T. mesenterina*, particularly concerning the MQY, stood out among
the three species. Time exerted a substantial influence on MQY ($p < 0.001$), as observed
in both nighttime and daytime measurements. Notably, during the dark treatment period,
MQY was significantly lower than during the light treatment condition (Fig. 4C).

In the comparison of treatments, corals subjected to anoxia exhibited a significant
reduction in photosynthetic efficiency during the initial 12 h of treatment in all three species.
In contrast, corals under hypoxic conditions displayed a slower response compared to the
anoxia treatment. Notably, both groups of corals ultimately recovered by the end of the
experiment (72 h). These findings underscore the unique temporal and treatment-specific
dynamics influencing the photosynthetic performance of *T. mesenterina*.

## Symbiodiniaceae density and chlorophyll content

The photosynthetic symbiont and pigments density exhibited a lesser impact under low
oxygen conditions (hypoxia and anoxia) (Fig. 5). In *P. acuta* corals, both hypoxia and
anoxia treatments led to a significantly lower density of Symbiodiniaceae compared to the

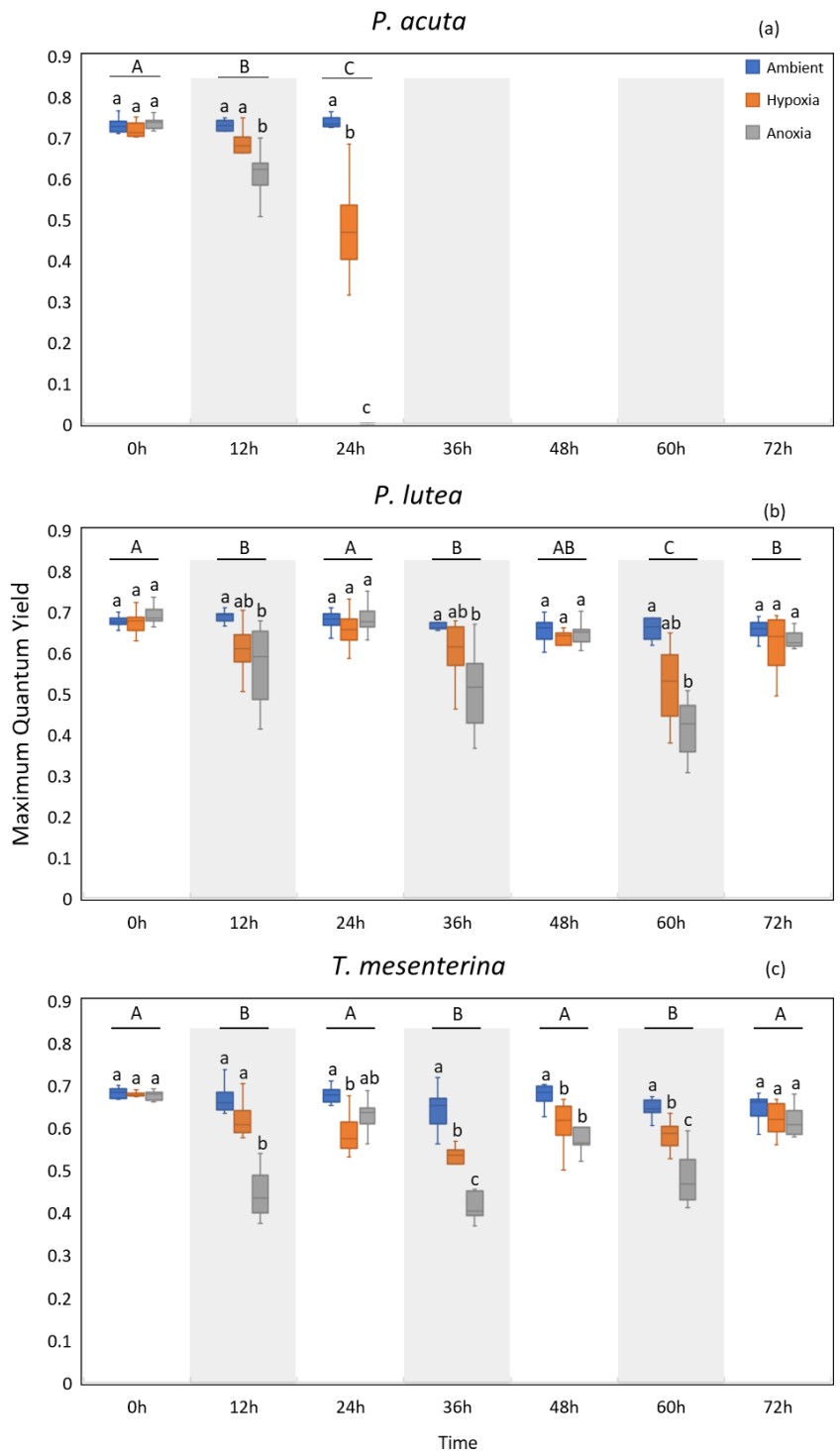

**Figure 4  Maximum quantum yield (MQY) of *P. acuta* (A), *P. lutea* (B), and *T. mesenterina* (C) under hypoxia and anoxia during a 72-h experiment.** The shaded area represents nighttime. Capital letters indicate differences between times. Lowercase letters denote differences between treatments.

initial group collected at the beginning of the experiment ($p = 0.049$, Fig. 5A), resulting in overall coral holobiont tissue loss. However, no significant difference in Symbiodiniaceae density was observed in *P. lutea* and *T. mesenterina* (Table S3). Additionally, anoxia treatment resulted in significantly lower concentrations of chlorophyll *a* (Table S4) and chlorophyll $c_2$ (Table S5), reflecting the impact of tissue loss. In *T. mesenterina*, hypoxia treatment led to a significant decrease in chlorophyll *a* concentration ($p = 0.014$). Notably, *P. lutea* showed no discernible effect from hypoxia and anoxia in these three parameters.

## Respiration and primary production
### Respiration
Hypoxia and anoxia conditions exerted significant impacts on respiration of all the species (Fig. 6, Table S6). The effects of hypoxia and anoxia were evident, as statistically significant decreases in coral respiration were observed ($p < 0.001$). However, there were no significant differences observed between the days of sampling for *P. lutea* and *T. mesenterina* ($p = 0.133$ and $p = 0.469$, respectively).

### Primary production
While *P. acuta* exhibited no significant effect on net primary production rates under hypoxia and anoxia ($p = 0.053$, Fig. 7A), *P. lutea* displayed variability in net primary production. Specifically, under hypoxic conditions, *P. lutea* showed no difference in net primary production. However, under anoxic conditions, it exhibited a significantly higher net primary production rate compared to other treatments across all three days ($p < 0.001$, Fig. 7B). Importantly, there was no discernible impact from the days of incubation, and no interaction effects were observed ($p = 0.849$ and $p = 0.876$, respectively). On the other hand, an interaction effect between treatment and day was observed in *T. mesenterina* ($p = 0.034$, Fig. 7C). The net primary production rates of *T. mesenterina* were significantly affected by hypoxia and anoxia ($p < 0.001$). Furthermore, under anoxic conditions, *T. mesenterina* exhibited significantly higher net primary production rates from the first day (Table S7).

In terms of gross primary production, *P. acuta* exhibited a significant reduction in the gross primary production rates under both hypoxia and anoxia conditions ($p = 0.001$, Fig. 8A). In contrast, *P. lutea* showed no significantly difference between treatments and days (Table S8), only displayed a lower gross primary production rate on the first day of the experiment (Fig. 8B). Unlike *P. acuta* and *P. lutea*, *T. mesenterina* showed significantly lower gross primary production rates specifically under anoxia (Fig. 8C) on the third day of the experiment ($p = 0.016$).

### Calcification
The calcification rate of *P. acuta* was significantly affected by anoxia ($p = 0.020$, Fig. 9A). In the case of *P. lutea*, calcification rates were influenced by both hypoxia and anoxia ($p < 0.001$, Fig. 9B). A significantly lower calcification rate under hypoxia was observed on the second day, while under anoxia, the calcification rate of *P. lutea* exhibited an effect across all three days. Calcification rates of *T. mesenterina* were significantly influenced by treatment ($p < 0.001$, Fig. 9C), day ($p < 0.01$, Fig. 9C), and with an interaction between
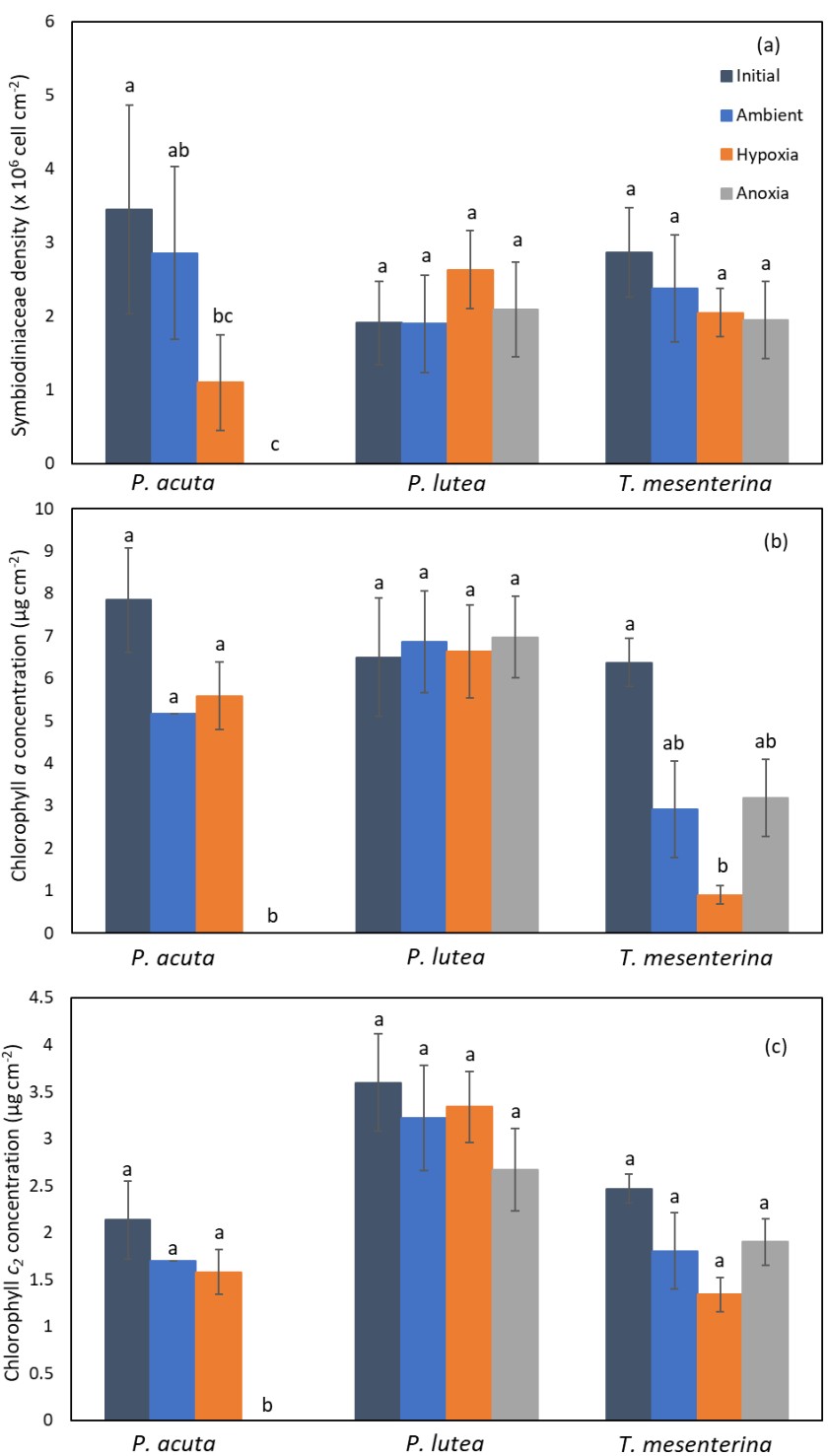

**Figure 5  Comparison of Symbiodiniaceae density (A), chlorophyll *a* concentration (B), and chlorophyll *c*$_2$ concentration (C) in *P. acuta*, *P. lutea*, and *T. mesenterina*.** Lowercase letters indicate differences between treatments.

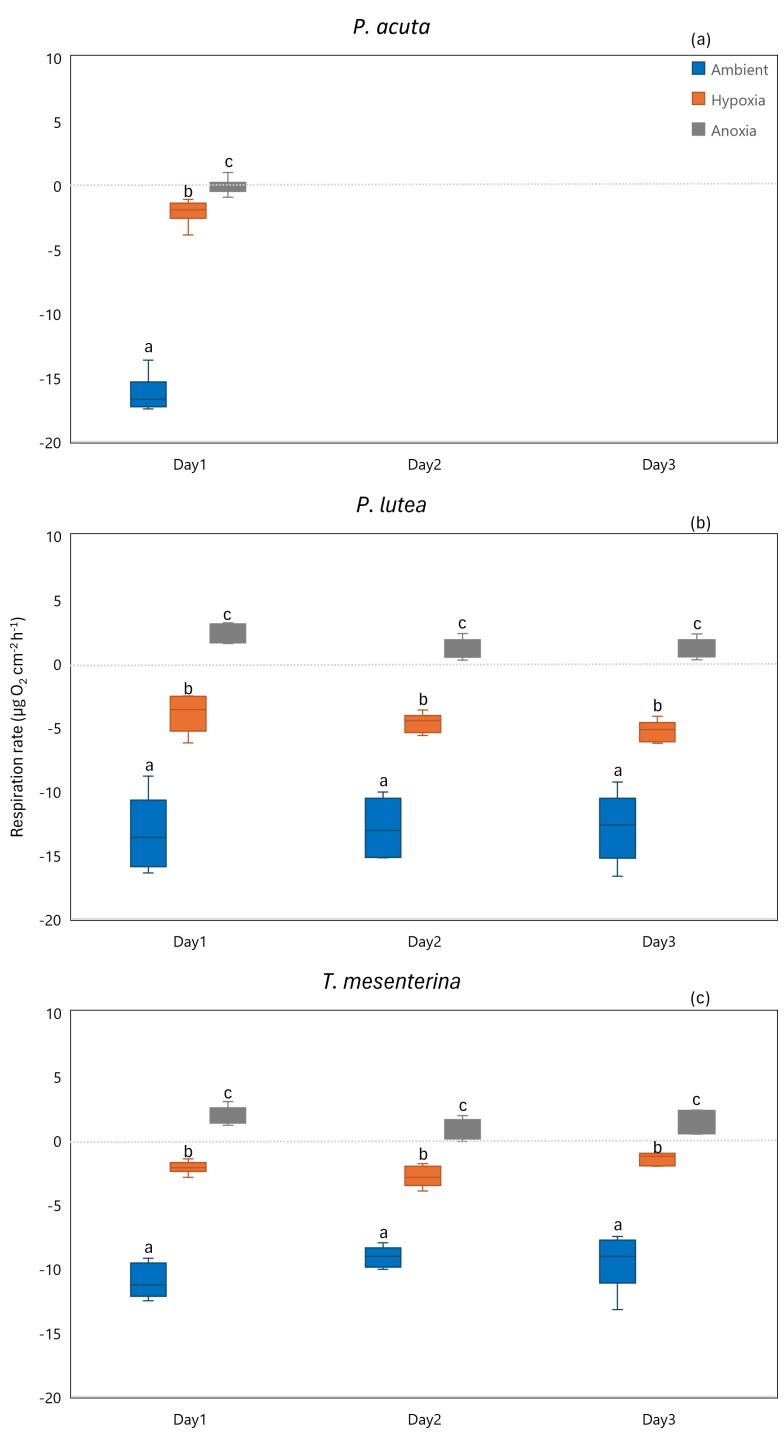

**Figure 6** **Respiration rates of *P. acuta*. (A), *P. lutea* (B), and *T. mesenterina* (C) from day 1 to day 3 under ambient, hypoxic, and anoxic conditions.** Lowercase letters denote differences between treatments, while uppercase letters indicate differences between days.

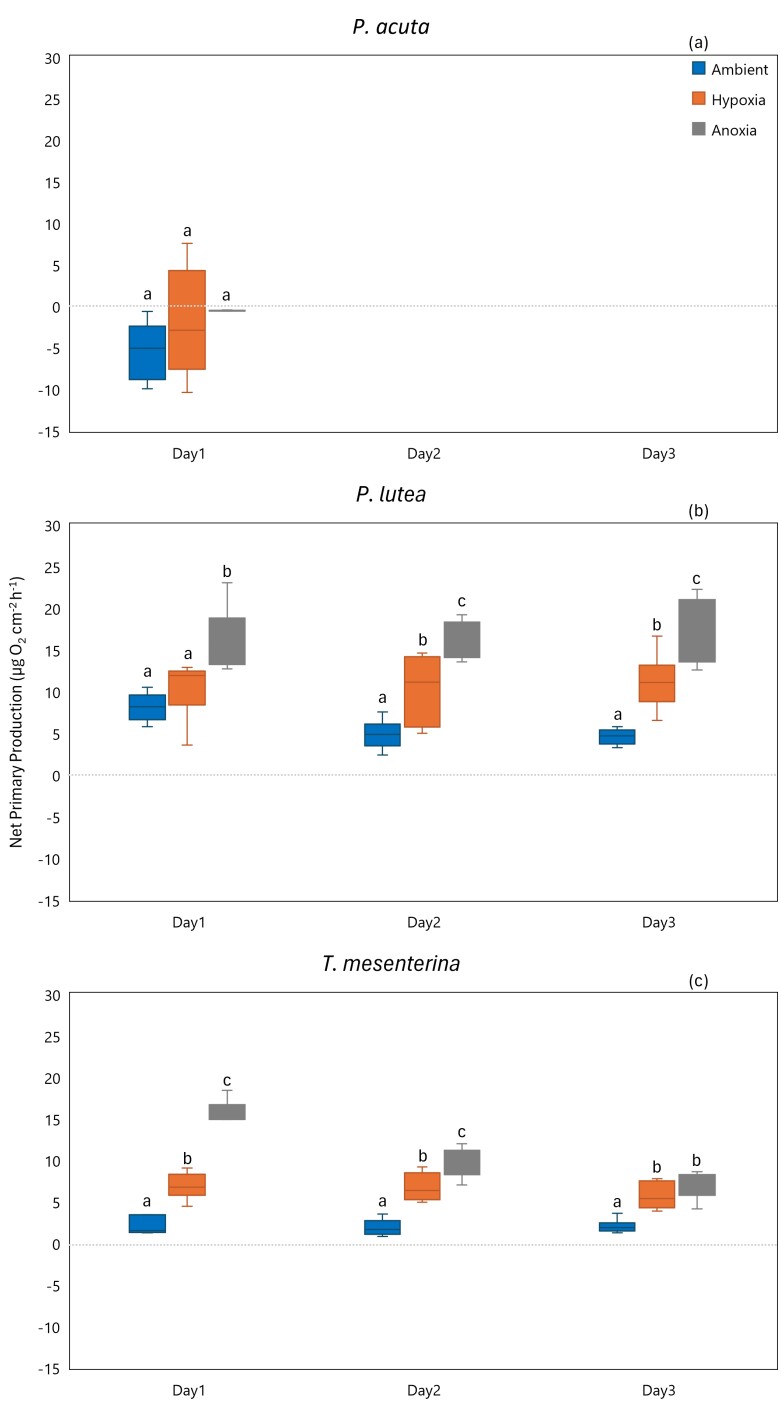

**Figure 7** **Net primary production from day 1 to day 3 of *P. acuta* (A), *P. lutea* (B), and *T. mesenterina* (C) under ambient, hypoxic, and anoxic conditions.** Lowercase letters denote differences between treatments, while uppercase letters indicate differences between days.

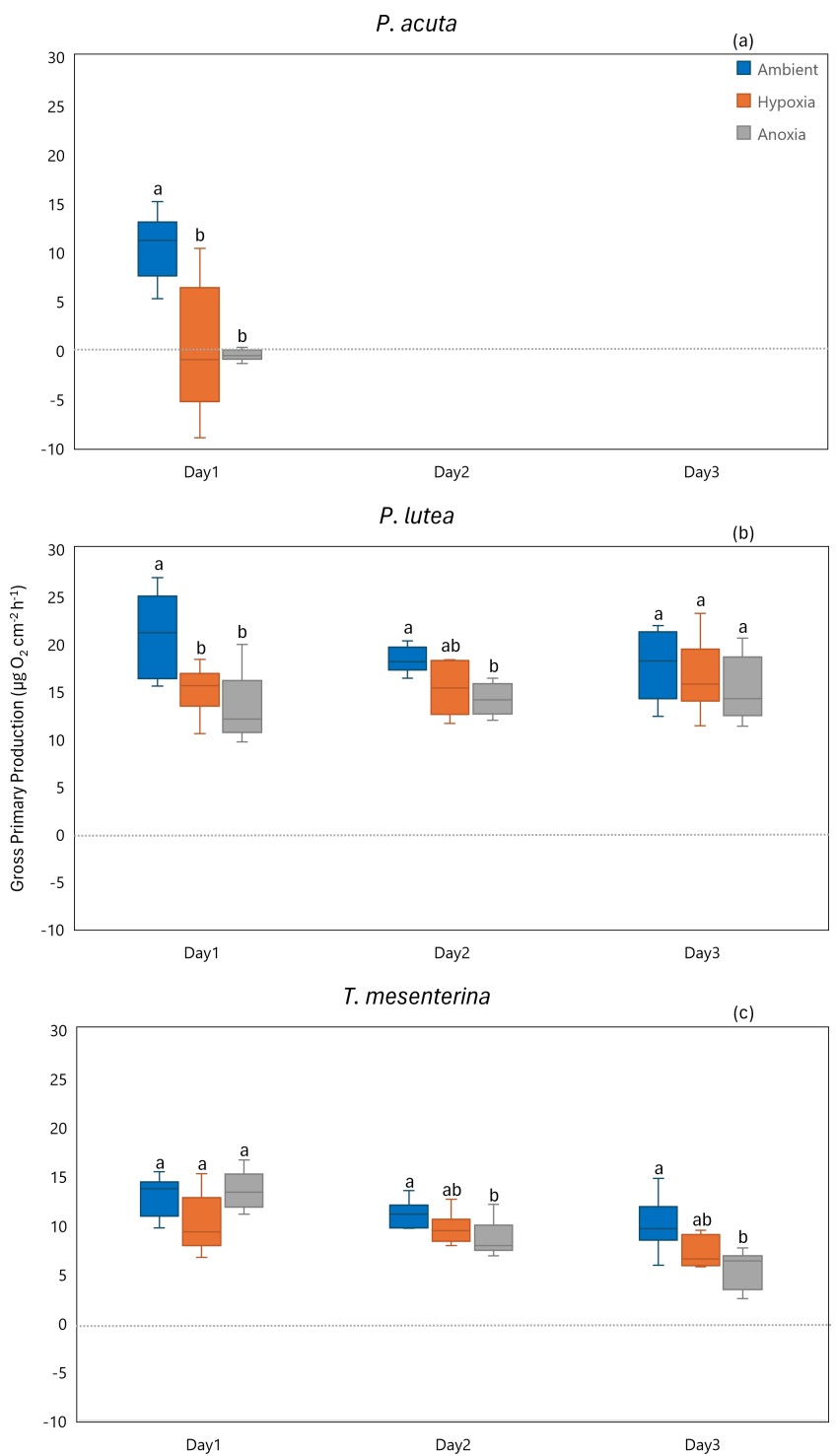

**Figure 8** **Gross primary production from day 1 to day 3 for *P. acuta* (A), *P. lutea* (B), and *T. mesenterina* (C) under ambient, hypoxic, and anoxic conditions.** Lowercase letters denote differences between treatments, while uppercase letters indicate differences between days.

these factors ($p < 0.001$, Table S9). Initially, on the first day of stress, the calcification rate of *T. mesenterina* remained unaffected by hypoxia or anoxia. However, a notable reduction was observed on the second and third days under these conditions.

## DISCUSSION

A comprehensive inquiry into the effects of oxygen limitation on coral ecosystems within the lower Gulf of Thailand is motivated by the urgent need to address the growing threat of deoxygenation events to coral health globally (*Altieri et al., 2017*; *Nelson & Altieri, 2019*; *Hughes et al., 2020*). Given the limited availability of DO data in Thailand, we initiated regular monitoring, revealing persistent hypoxic conditions in the study site (Fig. 3). Recorded data reveals a consistent decrease in seawater oxygen levels, reaching hypoxia (oxygen less than 2 mg $L^{-1}$), particularly during two periods (under red line): from the end of September to mid-October 2021 and in mid-May 2022, aligning with elevated temperatures (Fig. S1). The solubility of oxygen and metabolic requirements in aquatic ectotherms is intricately linked to water temperature (*Roman et al., 2019*). Moreover, the proximity to mainland of Kham Island, situated just 2 km from the estuarine and close to the coastal area (Fig. 1), makes it susceptible to anthropogenic activities and freshwater runoff, contributing to the phenomenon of ocean deoxygenation (*Laffoley & Baxter, 2019*; *Mancini et al., 2023*).

Based on these data, our objective was to investigate the impact of hypoxic and anoxic conditions on the physiological performance of the most common coral species, *P. acuta*, *P. lutea*, and *T. mesenterina* at Kham Island. Our findings reveal that diminished oxygen levels significantly influence various physiological processes, reducing the efficiency of photosystem II, and decreasing respiration, primary production, and calcification rates. Importantly, the observed effects are contingent upon the specific oxygen levels (hypoxia or anoxia) and associated with the morphological variations among different coral species.

Our investigation highlights a significant impact of low oxygen conditions on the photosynthetic performance of the three coral species. The decline in photosynthetic efficiency is typically associated to a structural alteration in photochemical reaction centers and/or the donor and acceptor sides and a reduction in photosystem II density, affecting electron transport, as corroborated by previous studies (*Gorbunov et al., 2001*; *Hill et al., 2004*; *Smith, Suggett & Baker, 2005*; *Duarte et al., 2017*; *Franzitta et al., 2020*; *Deleja et al., 2022*; *Smythers et al., 2023*). However, the recent findings of *Deleja et al. (2022)* propose that although photochemical reaction centers remain unchanged during nighttime hypoxia, there is an observed modification in the connectivity between the PSII antennae. This alteration results in a reduced absorption of the photon flux by the pigment antenna, ultimately leading to an insufficient amount of transported energy to the reaction centers (*Strasser & Stirbet, 2001*; *Duarte et al., 2015*). Furthermore, this photoinhibition may arise from oxidative stress (*Deleja et al., 2022*), impeding repair of the photosystem and exacerbating damage. Our findings indicate that the effects on photosynthesis related parameters were primarily observed in the MQY and, to a certain extent, in the gross photosynthesis rates. However, there was no significant impact on Symbiodiniaceae density

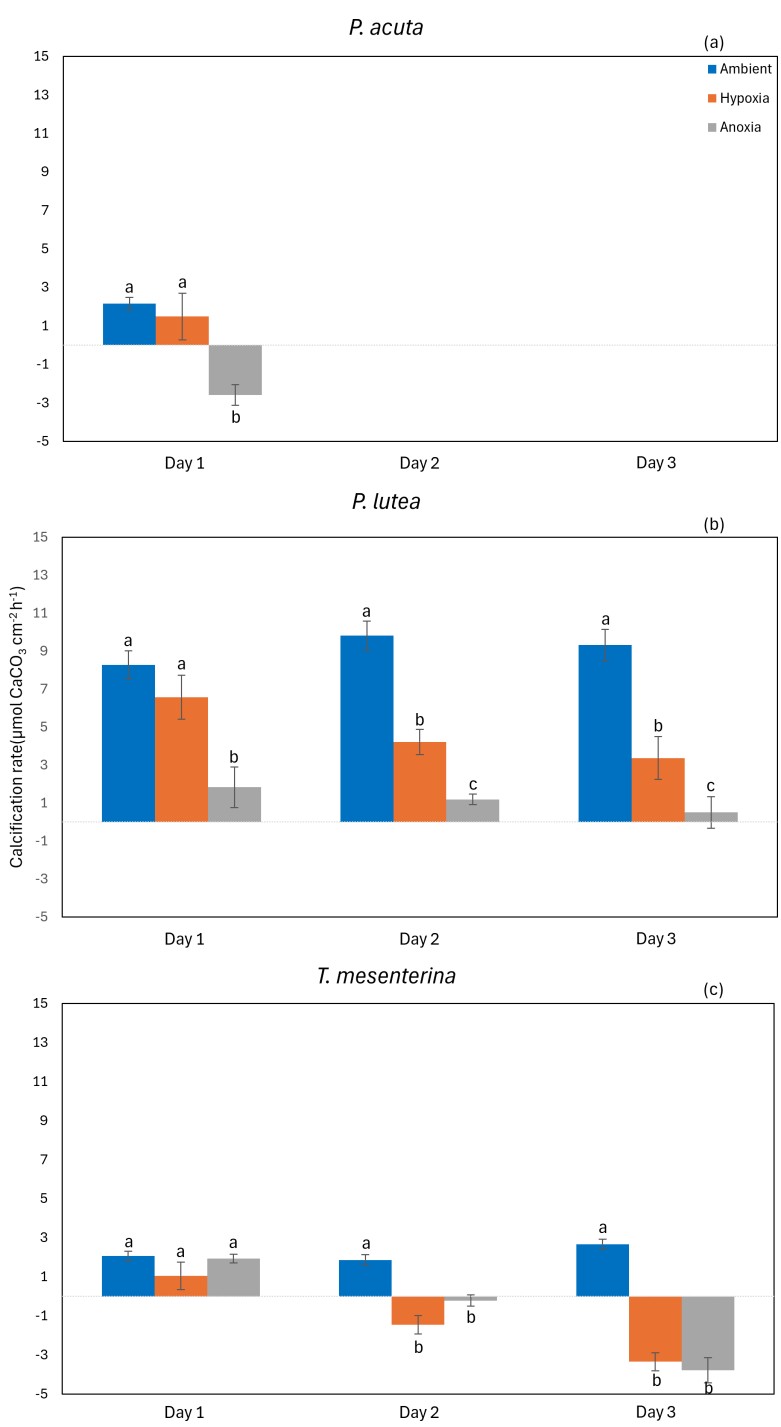

**Figure 9** Calcification rate from day 1 to day 3 of *P. acuta* (A), *P. lutea* (B), and *T. mesenterina* (c) under ambient, hypoxic, and anoxic conditions. Lowercase letters denote differences between treatments, while uppercase letters indicate differences between days.

or chlorophyll content. This finding aligns with our earlier research (*Jain et al., 2023*) and is consistent with findings from other studies (*Alva García et al., 2022*; *Deleja et al., 2022*). It suggests that chlorophyll fluorescence parameters may serve as effective biomarkers because of its susceptibility to sublethal stress (*Hoadley et al., 2022*) for detecting and assisting in the early identification of hypoxic and anoxic stresses in *P. acuta, P. lutea*, and *T. mesenterina*. Moreover, it is noteworthy that the impacts of hypoxia exhibited significant variations between periods with and without light, corresponding to daytime and nighttime conditions. This pattern may be attributed to the ongoing photosynthetic activity during the day, contributing to oxygen production and alleviating tissue oxygen levels. In contrast, during nighttime, when photosynthetic activity ceases and respiration consumes oxygen, the impact of oxygen deprivation intensifies.

Cellular respiration, a vital process for generating energy for cellular functions, was significantly impacted by low oxygen in our study. Under these conditions, *P. acuta*, *P. lutea*, and *T. mesenterina* exhibited reduced respiration rates. The observed influence of oxygen limitation on coral respiration aligns with findings from previous studies (*Dodds et al., 2007*; *Nelson & Altieri, 2019*; *Alva García et al., 2022*; *Gravinese et al., 2022*). Certain cnidarians have demonstrated the ability to tolerate acute hypoxic and anoxic conditions by transitioning from aerobic respiration to the less efficient anaerobic respiration pathway, enabling them to survive extended exposure periods (*Martinez, Smith & Richmond, 2012*; *Murphy & Richmond, 2016*; *Gravinese et al., 2022*). Consequently, the decrease in respiratory oxygen consumption during hypoxic and anoxic stress observed in our study may be linked to a gradual shift towards anaerobic respiration (*Nelson & Altieri, 2019*; *Gravinese et al., 2022*). However, this metabolic shift comes at the cost of energy production and may lead to an energy deficit stage (*Murphy & Richmond, 2016*). While our measurements were conducted using the holobionts, it is crucial to acknowledge the tight coupling of coral respiration with the photosynthesis of symbiotic algae. Previous studies have demonstrated that the carbon dioxide necessary for photosynthesis by Symbiodiniaceae is derived from coral cellular respiration (*Muscatine, Porter & Kaplan, 1989*). Consequently, low oxygen not only limits coral and Symbiodiniaceae respiration but also indirectly inhibits symbiotic algal photosynthesis by restricting the supply of carbon dioxide from coral respiration (*Harland & Davies, 1995*; *Gardella & Edmunds, 1999*). Despite this, it was found that photorespiration in *Galaxea fascicularis* remained unaffected by ambient oxygen levels even in a 20% situation (*Osinga et al., 2017*). This finding is consistent with observations in *P. lutea* under hypoxic conditions (Fig. 7B), suggesting that the impact of oxygen on coral photosynthesis varies depending on the species. It's important to note that, considering the lower MQY and gross photosynthetic rates, any increase in net primary production should not be interpreted as a positive impact for corals.

Calcification, an essential process for coral growth, exhibited high sensitivity to low oxygen levels in our study. Under these conditions, *P. acuta*, *P. lutea*, and *T. mesenterina* displayed reduced calcification rates. Coral calcification, a "photosynthesis-driven" process, relies on the energy derived from the photosynthesis of Symbiodiniaceae. Moreover, food supply increases the growth rates of coral, further supporting the calcification process.
This intricate interplay between photosynthesis and food availability underscores the importance of both factors in sustaining coral growth and reef health (*Rodolfo-Metalpa et al., 2008*; *Colombo-Pallotta, Rodríguez-Román & Iglesias-Prieto, 2010*). Additionally, the products of photosynthesis play a crucial role in fueling corals' aerobic respiration and the deposition of calcium carbonate, which demands a significant portion (13–30%) of the total metabolic energy budget from corals through aerobic respiration (*Cohen & Holcomb, 2009*; *Colombo-Pallotta, Rodríguez-Román & Iglesias-Prieto, 2010*; *Allemand et al., 2011*). Hence, the availability of oxygen plays a vital role not only in respiration and photosynthesis but also in restricting calcification by influencing both respiration and photosynthesis (*Colombo-Pallotta, Rodríguez-Román & Iglesias-Prieto, 2010*; *Wijgerde et al., 2012*). Previous studies consistently highlight the significant impact of oxygen on corals' calcification in both dark and light conditions (*Al-Horani, Tambutté & Allemand, 2007*; *Colombo-Pallotta, Rodríguez-Román & Iglesias-Prieto, 2010*; *Wijgerde et al., 2012*; *Wijgerde et al., 2014*; *Nakamura, Nadaoka & Watanabe, 2013*; *Zhang et al., 2023*). These findings underscore the sensitivity of coral calcification to oxygen levels, emphasizing potential implications for overall coral health and growth. Notably, observations of *Pocillopora* species' growth rates showed a substantial 43.3% reduction at relatively lower oxygen levels, as reported by *Castrillón-Cifuentes, Zapata & Wild (2023)*.

Considerable differences in responses among morphologically distinct corals have been documented, with branching and solitary coral colonies being more susceptible to severe hypoxic conditions compared to massive, sub-massive, and encrusting corals (*Guzmán et al., 1990*; *Simpson, Cary & Masini, 1993*; *Adjeroud, Andréfouët & Payri, 2001*). The skeletal porosity of corals is a critical trait linked to their environmental adaptability. Our research reveals that while *P. lutea* identifies as a perforate species, it distinguishes itself from *P. acuta* as imperforate. Unlike the branching *P. acuta*, the larger *P. lutea* features extensively perforated skeletons, allowing coral tissues to penetrate to greater depths. In the case of perforate *versus* imperforate corals, the internal environment dictates the effective three-dimensional habitat for Symbiodiniaceae, leading to either scattered or confined spatial configurations of symbionts within the coral host. Consequently, the morphotypes under study exhibit distinct relative partitioning of biological complexity: branching imperforate corals demonstrate high levels of spatial diversity and external structural complexity, whereas massive and foliose perforate corals exhibit simpler external structures but higher levels of internal complexity. This indicates inherent biological variability among corals, with perforate skeletons forming intricate, three-dimensional interior habitats that influence symbiotic dynamics (*Yost et al., 2013*). Due to their thicker tissues, large and encrusting corals like *Porites* possess greater energy reserves and photo-protective capabilities, rendering them more resilient to environmental stress. Interestingly, there appears to be a correlation between symbiont concentration in localized tissue areas and tissue thickness (*Qin et al., 2020*).

In our study, *P. acuta* exhibited the highest sensitivity to anoxic conditions, resulting in tissue loss and mortality within 24 h. This response mirrored the findings in the branching coral *Acropora cervicornis*, which also experienced tissue loss and mortality within a day of exposure to DO levels of 1.0 mg L$^{-1}$ (*Johnson et al., 2021*). These observations underscore

the vulnerability of branching corals to low oxygen stress. Given the susceptibility of *P. acuta* and similar branching species, it is crucial that management efforts prioritize these corals when developing conservation plans and strategies. Additionally, regional areas with high branching coral coverage should be closely monitored. Strict control measures should be implemented to mitigate factors that may cause algal blooms, and continuous monitoring of water body health and coral conditions is essential. This holistic approach will help ensure the long-term resilience and sustainability of coral reef ecosystems. In contrast, *P. lutea* and *T. mesenterina* did not show any mortality over a 72-hour period, indicating a higher tolerance to hypoxia, consistent with previous records for massive corals such as *P. lutea* and *Orbicella faveolata* (*Johnson et al., 2021*; *Alderdice et al., 2022*). This difference in tolerance by morphology also aligns with our recent findings from the Andaman Coast (*Jain et al., 2023*). However, the variability in hypoxia thresholds is not only evident among genera but also among coral species. For example, within the same genus and morphology, *Acropora selago* and *Acropora yongei* exhibited bleaching under hypoxic conditions within 12 h, while *Acropora tenuis* showed no bleaching under the same stress (*Haas et al., 2014*; *Alderdice et al., 2021*). Effective conservation efforts in the face of climate change should place importance on understanding the biology of corals, considering both the variation within and among species. Tailoring conservation strategies to specific coral species, especially those with distinct sensitivities to environmental stressors like hypoxia, is critical for the long-term health and resilience of coral reef ecosystems.

In this study, we discovered a notable correlation between ambient conditions and low oxygen levels. However, it's essential to acknowledge the limitations imposed by the closed system and water flow dynamics, which may have amplified the impact on coral photorespiration and growth (*Larkum, Koch & Kühl, 2003*; *Schutter et al., 2011*). Consequently, there is a clear need for additional monitoring and experiments to thoroughly investigate the effects of low-oxygen conditions on coral health and resilience.

## CONCLUSIONS

In summary, our experiments highlight the sensitivity of corals to hypoxic and anoxic conditions, impacting essential processes related to energy balance and photosynthetic efficiency. Variability in resilience was evident among species, with *P. acuta* identified as the most susceptible. This study emphasizes species specific variations in vulnerability, linked to different morphologies, under low oxygen conditions, corroborating the earlier suggestion that branching corals are more sensitive to stress.

Our research gains particular relevance considering the persistent hypoxia in the natural environment of our study site. As challenges related to oxygen availability intensify due to climate change and coastal pollution, the implications for coral ecosystems become increasingly significant. A comprehensive understanding of these physiological processes is not only crucial for predicting the consequences of deoxygenation, as well as of climate change in general, but also for developing effective strategies to assess and mitigate the impacts of deoxygenated events on tropical corals.

## ACKNOWLEDGEMENTS

We would like to thank the Coastal Oceanography and Climate Change Research Center (COCC) and the Marine and Coastal Resources Institute (MACORIN) for providing the instrument and material support as well as the staff support in the field and the laboratory: Muhammad Arif Samshuri, Verdiana Vellani, Thanyapat Chamnina, Wilawan Hwan-air, Natchanon Kiatkajornphan, Rattaporn Sengkhim, Natthaya Sotthiphan, Jittapon Patcharat, Watchara Samsuvan.

### Funding

The research and student grant were supported by Prince of Songkla University (Grant number: ENV6402039S) and the PSU-TUYF Charitable Trust Fund under a project "Coral reef biodiversity conservation and connectivity in southern Gulf of Thailand to support reef resilience and sustainable use". The funders had no role in study design, data collection and analysis, decision to publish, or preparation of the manuscript.

### Grant Disclosures

The following grant information was disclosed by the authors:
Prince of Songkla University: ENV6402039S.
PSU-TUYF Charitable Trust Fund.

### Competing Interests

The authors declare there are no competing interests.

### Author Contributions

- Ying Long performed the experiments, analyzed the data, prepared figures and/or tables, authored or reviewed drafts of the article, and approved the final draft.
- Sutinee Sinutok conceived and designed the experiments, analyzed the data, authored or reviewed drafts of the article, facility and field work instrument, and approved the final draft.
- Pimchanok Buapet conceived and designed the experiments, analyzed the data, authored or reviewed drafts of the article, facility and lab work instrument, and approved the final draft.
- Mathinee Yucharoen conceived and designed the experiments, performed the experiments, analyzed the data, prepared figures and/or tables, authored or reviewed drafts of the article, facility and funding security, and approved the final draft.

### Ethics

The following information was supplied relating to ethical approvals (i.e., approving body and any reference numbers):

The experiment of this study was conducted according to the Animals for Scientific Purposes Act, B.E. 2558 (A.D. 2015) and approved by Institutional Animal Care and Use Committee, Prince of Songkla University (ref.46/2021).

## Field Study Permissions

The following information was supplied relating to field study approvals (i.e., approving body and any reference numbers):

The research permission in the Non-Hunting Area was approved by the Department of National Parks, Wildlife and Plant Conservation (permission number: 21685). Coral collection was permitted by the Department of Fisheries, Ministry of Agriculture and Cooperatives (permission number: 409) under Wild Animal Conservation and Protection Act, B.E. 2562 (A.D. 2019). The experiment of this study was conducted according to the Animals for Scientific Purposes Act, B.E. 2558 (A.D. 2015).

## Data Availability

The raw measurements are available in the Supplementary File.

## Supplemental Information

Supplemental information for this article can be found online at http://dx.doi.org/10.7717/peerj.18095#supplemental-information.

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
