# Peer review of "Unraveling the physiological responses of morphologically distinct corals to low oxygen"

_PeerJ, doi:10.7717/peerj.18095_

## Round 0.1 · original submission · Major Revisions

Thank you for your submission and for doing this important work in an understudied region. We appreciate your efforts on this work.

Three outstanding experts in the field have agreed to review your manuscript and for this we are grateful. Their reviews point to significant shortcomings which we truly hope you are able to address carefully and thoroughly in a revision. In particular, there is a lack of clarity on the experimental design which can have major impacts on the appropriateness of the statistical methods used: repeated sampling of the same individuals must be accounted for using proper methods for non-independence of samples through time (we recommend repeated measures or similar mixed-model - random and fixed effect - designs).

The reviewers have offered extensive edits and suggestions to improve the manuscript, and I expect that a revision would include careful accounting of every suggestion and how it was incorporated or otherwise addressed in the manuscript so that the reviewers and I have the easiest time verifying the improvements and clarifications.

Reviewer 1 ·

Basic reporting

No comment

Experimental design

No comment

Validity of the findings

No comment

Additional comments

See the attached PDF

Annotated reviews are not available for download in order to protect the identity of reviewers who chose to remain anonymous.

·

Basic reporting

Background information is sufficient and provides rationale for the proposed study. In line 97, it is necessary to spell out the abbreviated species names for clarity. Additionally, a citation is needed to support the assertion that "physiological oxygen requirements for many organisms increase." This citation would provide scientific backing to the claim, enhancing the credibility of the statement and ensuring that the information is properly sourced.

Experimental design

The research question is clearly defined, however, more information is required in the methodology section to understand the statistical approach and validity of the research findings.
- To enhance the justification for selecting the three species, it would be beneficial to provide additional details on the species utilized, their prevalence and population density at the specific site, as well as the rationale behind their selection. Incorporating this information into the methodology section would bolster the rationale behind the choice of these species.
- A statement that the corals recovered from the fragmenting would be helpful to determine that a week of acclimation was long enough to see the tissue regeneration at the site of fracture.
- Include the units for the dissolved oxygen measurements, pH measurements (NSB, total, seawater scale?) and total alkalinity.
- I am happy to see that the authors included the specific parameters used for the PAM. This is extremely helpful with comparisons across studies.
- Can you provide evidence of the artificial seawater solution that was used (3.2% NaCl).
- Additional details regarding the experimental procedures, particularly concerning the chambers and incubations, are necessary for clarity. For instance, crucial information such as the size and volume of the chambers is absent. It is unclear whether 72 individual chambers were employed for each coral fragment and if mixing occurred within these chambers. Furthermore, the methods for monitoring conditions within each chamber and whether the chambers were fully closed need clarification. While the volume of the chamber is integral to the equation for calculating Net Primary Productivity (NPP), it is not explicitly stated in the methods section. Additionally, the biological mass-to-volume ratio holds significant importance for determining respiration and photosynthetic rates, as highlighted in Svendesen et al. (2016). Lastly, it is unclear whether NPP, respiration (R), and Gross Primary Productivity (GPP) were standardized to coral surface area or mass.
o Svendsen, M. B. S., Bushnell, P. G., & Steffensen, J. F. (2016). Design and setup of intermittent‐flow respirometry system for aquatic organisms. Journal of fish biology, 88(1), 26-50.
- To enhance clarity, it is essential to address several key questions regarding the experimental procedure. Firstly, the duration of coral incubation needs clarification. Additionally, it's crucial to specify the time variable utilized in the calculation of calcification rates. Moreover, information regarding the volume of water used in the incubation process is lacking. If the corals are being incubated for 12 hours in the same volume of water, it raises concerns about potential issues such as the accumulation of waste materials, which could adversely impact the corals. This accumulation may confound the experimental treatment, leading to misleading results. Therefore, adjusting the duration or volume of incubation water may be necessary to mitigate these potential confounding factors and ensure the accuracy of the experimental outcomes.
- Experimental design: Was every genet or genotype of each species adequately represented within each treatment group?
- Regarding the oxygen sensor, could the authors provide additional details concerning the specific type of oxygen sensor utilized? Moreover, clarification on the frequency of dissolved oxygen (DO) measurements, along with whether they were conducted in the bulk seawater or in close proximity to the coral surface, would be beneficial.
- The statistical approach requires further clarification for better understanding. It remains unclear which factors were included in the two-way ANOVA analysis. Additionally, additional information regarding the factor "day" is necessary. If "day" represents the time of exposure and the measurements are taken repeatedly from the same individuals over time, a repeated measures design should be incorporated into the statistical approach to address the non-independent nature of these measurements. Consequently, more explicit elucidation on the statistical methodology is warranted, including details on how potential sources of variability were addressed, such as the repeated measures aspect.

Validity of the findings

It is difficult to determine the validity of the research findings without more information regarding the experiment setup, specifically the size of the chambers and length of incubation.
- In situ measurements of dissolved oxygen levels offer valuable background information regarding the typical conditions to which the corals are exposed. However, it remains unclear whether the corals were all collected from the same area where the loggers were deployed. Clarification on this point would enhance our understanding of the environmental context in which the corals were studied.
- Figure 2 indicates that chlorophyll and symbiont measurements were taken at 0 hours and 72 hours, while Figure 5 presents data labeled as "initial" and the various treatments. However, it remains ambiguous whether "initial" refers to measurements taken before exposure in each treatment or if it represents ambient conditions. To enhance clarity, it would be beneficial to specify whether "initial" denotes the pre-exposure measurements within each treatment. Additionally, considering the inherent biological variation among fragments, genotypes, and species, it might be advantageous to analyze the percent change before and after treatment. Standardizing each individual to its initial measurement could effectively control for biological variation and provide a clearer understanding of treatment effects.
- The field-based data clearly indicates that these corals frequently experience hypoxic conditions, suggesting that such conditions may serve as a selective pressure at the field site.
- In the discussion section, the authors address the variation in responses potentially attributed to morphological factors. However, the study does not explicitly mention morphological differences between the species until this point. Although the examined species exhibit distinct morphologies, other significant differences such as skeletal morphology, symbiont types, and tissue thickness are also likely to contribute to hypoxia sensitivity and merit discussion. This is particularly relevant given that the main finding and argument presented by the authors revolve around early impairment of the photosystems. Therefore, a comprehensive examination of all pertinent morphological and physiological differences among the species would enrich the discussion and provide a more nuanced understanding of the observed responses to hypoxia.

Additional comments

Improvement can be made to the manuscript by strengthening the connection between the figures and the in-text discussion. This can be achieved by utilizing consistent language in describing the treatments both in the figures and in the accompanying text. By aligning the terminology used to describe the treatments, readers can easily comprehend the relationship between the figures and the corresponding discussions, enhancing the overall coherence and clarity of the manuscript.

·

Basic reporting

In terms of writing, there are a number of statements within the interpretation of the results that require rephrasing for clarity or simply removed, but other than that the manuscript is relatively well-written. Literature on the topic of hypoxia and corals is relatively well-covered, however, there is an inappropriate use of a reference which needs correcting (refer to specific comments in annotated manuscript). Resolution of figures needs to be improved and there is a lot of white space in figures, Authors could consider rescaling. Raw data appears to be fully accessible in supplementary material and statistical results are also reported.

Experimental design

The manuscript of Long et al presents key physiological responses to low oxygen levels for 3 coral species which is an important complementary dataset to existing datasets (from other species/reef regions) to help understand the impact of ocean deoxygenation on corals and fits the scope of this journal. Methods presented are commonly used within coral physiological assessments. However, clarification or justification of certain important details is needed in the methods including why very low light levels were used, where exactly the dissolved oxygen levels were positioned on the reef, and whether differences in pH of the seawater post nitrogen bubbling was considered. Notably, an incorrect formula was used for gross respiration. Refer to the specific comments within the annotated manuscript pdf for all other details to clarify/justify.

Validity of the findings

Replication of n=8 colonies per treatment per species is sufficient and routine statistical testing is used and reported. However, some discussion points or conclusions need amended which are ambiguous or not directly supported by the data presented such as the impact of hypoxia structural integrity (of the photosystems) and the suggestion to use chlorophyll fluorescence as a biomarker for hypoxia.

Additional comments

Specific comments are provided in the attached annotated pdf of the manuscript and below for ease of author replies.

Specific comments:
Line 37, Is this statement also referring to anoxia or hypoxia?
Line 50, Clarify why you present a range of up to 3.5mg/L, and whether you are referring to hypoxic water conditions or hypoxia intratissue condition.
Line 56, For clarity, change '-' to 'and' so readers can follow which RCP results in which %
Line 61, Specify heat source and whether this is referring to ocean warming or atmospheric warming.
Line 84, Provide an alternative reference as this reference does not support this statement.
Line 86, Update this reference to the correct publication date of 2021.
Line 87, Provide reference for this sentence.
Line 95, Change to 'previous work'
Line 97, Provide full names when first mentioned.
Line 111, in terms of? Specify how the condition of the reef was determined.
Line 115, was the logger attached to a coral or? this is important information to specify for interpreting the recorded data.
Line 116, reformat to 22nd June, correct throughout paper.
Line 118, specify how often measurements were taken in main text.
Line 135, Light levels seem very low, please provide a justification.
Line 138, Were the corals tested for any health measurements e.g. photosynthetic efficiency prior to experiment?
Line 148, Was pH adjusted or measured after nitrogen bubbling?
Line 156, what size were the chambers?
Line 158, What was the extent of oxygen draw down? Provide the rate in order to help identify whether the water volume to sample size was appropriate for measurements.
Line 160, change to 'start and end'.
Line 170, change to 'acclimation'.
Line 173, how were these settings determined, provide reference where possible.
Line 182, clarify whether the supernatant or pellet was used for the cell counts. How many technical replicates were used? Report in main text rather than only supplementary data.
Line 185, was ‘remaining slurry’ the algal pellet again? please specify.
Line 210, is GPP not calculated by adding respiration to NPP? Please check, cite appropriate reference and correct throughout results/paper.
Line 232, Not sure what is meant by this sentence? do you mean one-way anova was used to test for statistical significance of measurements prior to the death of this species?
Line 243, previously mentioned that a few months were not captured? please clarify.
Line 244, justify why 12:00 and 14:00 sample timings were used.
Line 252, Was hypoxia identified during the 12:00 or 24:00 (or both) measurement?
Line 266, were the reductions in hypoxia and anoxia relatively the same for the same species? i.e. could the hypoxia and anoxia response be ranked similarly for each species?
Line 267, assuming this was the case for only Porites?
Line 277, in terms of all coral species? or only T. mesenterina?
Line 290, Is 'density' missing from the sentence? Symbiont density?
Line 293, change to 'resulting in overall coral holobiont tissue loss'
Line 310, specify what was compared for the interactions.
Line 364, This is great to hear - will the monitoring continue? Please provide the duration of the hypoxia events in your dataset.
Line 375, do you mean 'most common.. within this region'?
Line 377, integrity or just efficiency?
Line 383, change to 'is typically associated' as you have not exactly shown this detail in this study.
Line 399, Arguably, an impact on the function of symbiotic algal photosynthetic efficiency will disrupt the symbiosis as the coral largely depends on photosynthates. Consider removing this statement.
Line 403, Provide more of an explanation on why Chl a fluorescence could serve as a biomarker and explain how other environmental conditions impact chl fluorescence.
Line 427, This is a good point to make
Line 429, Please clarify or remove this statement as it is not clear what you mean.
Line 436, What about heterotrophic feeding by corals, please address this aspect.
Line 451, Was there a significant difference in calcification rates between species? How was the calcification rates of Pocillopora in this study?
Line 463, General stress or hypoxia? please specify.
Line 464, If these species are most sensitive to stress, surely it would be better if these species were targeted for stress priming programs. Please provide more specific detail than 'conservation plans'.
Line 481, Justify the use of structural integrity as this study does not cover this metric but rather photosynthetic efficiency.

---

## Round 0.2 · Major Revisions

Apologies for the lengthy delays in the review process. Two of the original reviewers declined to review the revised version, and after a long search for replacements I stepped in and reviewed the manuscript myself. I think you have done a good job addressing the reviewer suggestions in this version, but there are a number of areas where you need to revise to make the manuscript suitable for publication. Most I think will be fairly straightforward to address, but it is crucial that you properly run the repeated measures models recommended by R2 correctly. I have provided additional suggestions as comments on your Response to Reviewers attached, and if you can return a summary of how you have addressed my comments I would be happy to finalize the acceptance. I also want you to add some deeper analysis of the increased NPP under anoxic conditions...one reviewer asked for discussion of photorespiration, but Please note that the increased NPP in anoxic conditions should be interpreted with caution because it is a larger proportional change in oxygen which may increase flux and diffusivity (0.5mg/L DO release in anoxic conditions may be equivalent to just 0.2 mg/L DO release in oxic conditions because of increased diffusivity in anoxia). Please do what you can to read up on this pattern and address it if possible.

Reviewer 1 ·

Basic reporting

no comment

Experimental design

no comment

Validity of the findings

no comment

Additional comments

I'd like to thank the authors for addressing my comments.

---

## Round 0.3 · accepted · Accept

Thank you for your thorough and attentive work addressing the editorial comments. We are pleased to accept this for publication!